# Epistemic Alignment: A Mediating Framework for User-LLM Knowledge Delivery

**Nicholas Clark, Hua Shen, Bill Howe & Tanushree Mitra**
Information School
University of Washington
Seattle, WA 98105, USA
{nclark4,huashen,billhowe,tmitra}@uw.edu

## Abstract

Large Language Models (LLMs) increasingly serve as tools for knowledge acquisition, yet users cannot effectively specify how they want information presented. When users request that LLMs "cite reputable sources," "express appropriate uncertainty," or "include multiple perspectives," they discover that current interfaces provide no structured way to articulate these preferences. The result is prompt sharing folklore: community-specific copied prompts passed through trust relationships rather than based on measured efficacy. We propose the *Epistemic Alignment Framework*, a set of ten challenges in knowledge transmission derived from the philosophical literature of epistemology, concerning issues such as uncertainty expression, evidence quality assessment, and calibration of testimonial reliance. The framework serves as a structured intermediary between user needs and system capabilities, creating a common vocabulary to bridge the gap between what users want and what systems deliver. Through a thematic analysis of custom prompts and personalization strategies shared on online communities where these issues are actively discussed, we find users develop elaborate workarounds to address each of the challenges. We then apply our framework to two prominent model providers, OpenAI and Anthropic, through structured content analysis of their documented policies and product features. Our analysis shows that while these providers have partially addressed the challenges we identified, they fail to establish adequate mechanisms for specifying epistemic preferences, lack transparency about how preferences are implemented, and offer no verification tools to confirm whether preferences were followed. For AI developers, the Epistemic Alignment Framework offers concrete guidance for supporting diverse ways of knowing; for users, it works toward information delivery that aligns with their specific needs rather than defaulting to one-size-fits-all approaches.

## 1 Introduction

Large Language Models (LLMs) have emerged as powerful knowledge tools, yet their flexibility raises the question of how to ensure they deliver information in a way that matches individual preferences about knowledge quality, evidence standards, and perspective diversity. While technical advances have proposed mitigations for hallucination (Ji et al., 2023; Shi et al., 2024; Mishra et al., 2024; Orgad et al., 2025) and uncertainty expression (Yona et al., 2024; Mohri & Hashimoto, 2024), a more subtle problem persists: **the misalignment between how users want knowledge presented and the limited mechanisms available to express these preferences**. For example, when a medical researcher requests "recent peer-reviewed sources," or a policy analyst seeks "balanced representation of competing viewpoints," they encounter interfaces that reduce these rich requirements to unstructured natural language instructions with inconsistent interpretation and no verification mechanisms.

Drawing on the theories of social epistemology and epistemic cognition, we formalize this misalignment as the *epistemic alignment problem*, and offer four contributions toward

understanding this challenge. We (1) introduce a framework for evaluating how well systems accommodate user preferences about knowledge delivery, (2) validate our framework through a thematic analysis of user attempts to control knowledge delivery with prompting strategies shared on online platforms, (3) assess current systems against this framework to identify specific interface limitations, and (4) consider requisite interface features that enable users to express and verify their preferences about how information should be presented, sourced, and qualified. Our work suggests that addressing the epistemic alignment problem requires rethinking how users communicate knowledge preferences to LLM-based systems, shifting from imprecise natural language instructions to structured interfaces that support explicit specification of parameters and provide transparent feedback about how these parameters shape knowledge delivery.

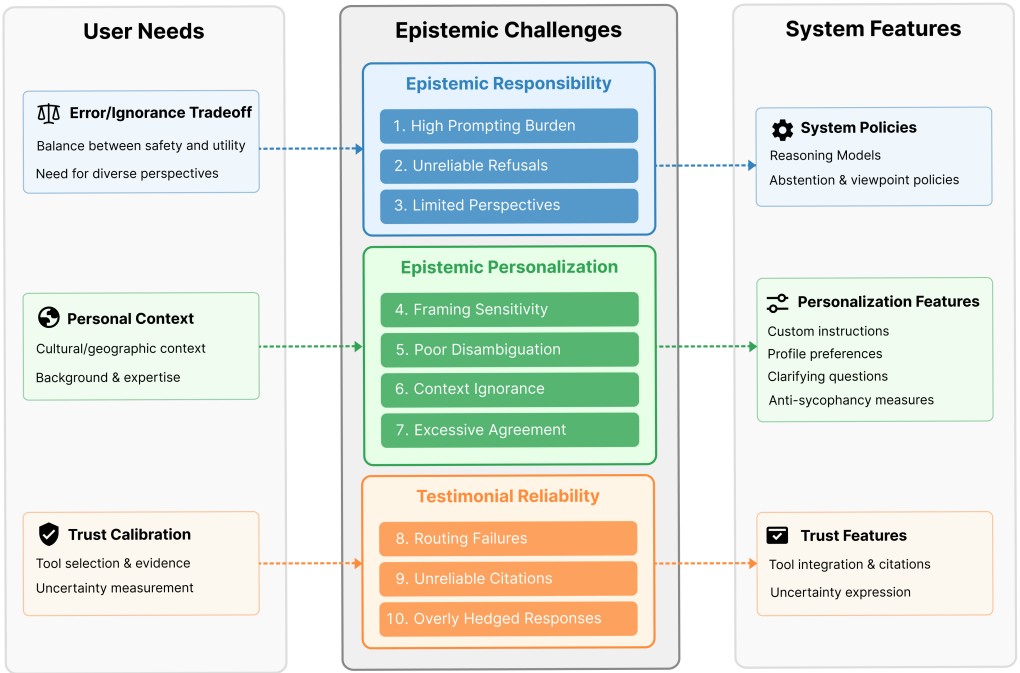

Figure 1: **The Epistemic Alignment Framework as a mediating structure between user needs and system implementation.** The framework identifies ten challenges across three epistemic dimensions: Epistemic Responsibility (challenges 1-3), Epistemic Personalization (challenges 4-7), and Testimonial Reliability (challenges 8-10). The framework serves as an intermediary layer for evaluating how well systems accommodate diverse epistemic preferences and identifying areas where current interfaces fail to support effective knowledge delivery. Each challenge was selected based on two criteria: theoretical grounding in epistemology and empirical validation through prevalence in user custom instructions.

## 2 Related Work

### 2.1 Philosophical Foundations for Epistemic Alignment

We draw from literature in epistemology, the philosophical subarea concerned with knowledge creation and transmission, and epistemic cognition, a topic in educational psychology relating to how people conceptualize knowledge and its acquisition. In particular, we rely on prior work in inquiry and social epistemology that considers how someone ought to responsibly engage with technology for knowledge-related activities.

**Inquiry** The object of our epistemic activities is *inquiry* (Hookway, 1994), the self-directed process through which we ascertain knowledge. The goal of inquiry inevitably varies

depending on the circumstance. For instance, sometimes we desire a deep, nuanced understanding of an issue; at other times we may be satisfied with a cursory familiarity. The primary vehicle through which we conduct inquiry is by posing questions (Hookway, 2008). The ultimate success or failure of an intellectual investigation in large part relies on the selection and quality of questions (Watson, 2018).

Performing inquiry in the digital age presents challenges, as the large volume of information mediated by opaque discovery mechanisms, such as web search and recommender systems, may give rise to *illusions of understanding*, where users believe they conducted a thorough investigation when, in fact, their methods are shallow or imperfect (de Ridder, 2022). We consider how these concerns arise when conducting inquiry with LLMs.

**Inquisitive Meta-Cognitive Tasks** To combat illusions of understanding, de Riddler, drawing from Hookway, formulates a set of meta-cognitive tasks requisite to conducting good inquiry (de Ridder, 2022; Hookway, 2003): 1) posing good questions or identifying good problems, 2) identifying good strategies for carrying out inquiries, 3) recognizing when we possess an answer to our question or a solution to our problem, 4) assessing evidence quality for some proposition, and 5) judging when we have considered all or most relevant lines of investigation. These meta-cognitive tasks establish clear criteria for effective inquiry, but in practice, users employ diverse strategies when executing each task, from choosing which questions to pursue to determining when evidence is sufficient. The selected strategies often reflect some combination of practical constraints and personal preferences. We consider user needs when interacting with LLMs for each meta-cognitive task to ensure complete coverage of the inquiry process.

**Epistemic Cognition** The topic of epistemic cognition (Greene et al., 2016) helps explain this variation in inquiry strategies by revealing the connections between beliefs about knowledge and methodology choices. In particular, the AIR framework decomposes the personal epistemology of an individual into their **A**ims, **I**deals, and **R**eliable processes (Chinn & Rinehart, 2016). Individual assumptions about what constitutes knowledge and how it can be verified directly shape learning strategies, information seeking behaviors, and decision-making processes. These epistemic beliefs vary across cultures (Chan & Elliott, 2004) and disciplines (Hofer, 2000), explaining why users might employ radically different approaches for the same meta-cognitive task. We contend that users bring similarly diverse strategies and requirements when engaging with LLMs.

## 2.2 Technical Approaches for Knowledge Delivery in LLMs

While philosophy provides the theoretical grounding for understanding epistemic challenges, recent technical work has begun addressing aspects of knowledge delivery in LLMs.

**Knowledge Acquisition** Prior work on the use of LLMs for knowledge acquisition has centered on particular failure modes such as knowledge conflict (Xu et al., 2024; Wang et al., 2024; Jin et al., 2024), hallucinations (Huang et al., 2025), factuality (Wang et al., 2023), or, on techniques to facilitate knowledge delivery such as citations (Gao et al., 2023). However, prior research primarily assess LLMs' knowledge acquisition capability with task performance (e.g., hallucinations, factuality), but lack a systematic study of what humans demand to trust the AI-generated knowledge.

**User Intent** Beyond these technical assessments of knowledge quality, understanding how users communicate their epistemic needs presents its own set of challenges. In open-domain QA, Min et al. showed that real user questions are often ambiguous, proposing systems that enumerate every plausible answer and rewrite the query to clarify each interpretation. Other work attempts to infer hidden intent from the prompt itself: for example, Bodonhelyi et al. (2024) developed a fine-grained intent taxonomy and found that even advanced LLMs sometimes fail to correctly recognize specific user intents, leading to user dissatisfaction. However, these studies primarily examined user intent within general conversations or question-answering tasks, without addressing the specific factors that shape users' needs when evaluating and trusting AI-generated knowledge.

**Prompt Sharing**   Given these difficulties in expressing intent through natural language, users have turned to community-based strategies for sharing effective prompts and workarounds. Mahdavi Goloujeh et al. (2024) studied text-to-image generation with Midjourney, conducting a qualitative analysis of semi-structured interviews with members of the community. Shen et al. (2024b) analyzed jailbreak prompts for LLMs, collecting over 1,400 examples from communities across Reddit and Discord. Trippas et al. (2024) examined prompt behavior for Google Bard, analyzing prompt logs for features such as length and semantic diversity. However, while these studies use similar methods, Goloujeh focuses on a different modality, and neither Trippas nor Shen explicitly examine epistemic preferences.

## 3   Problem Definition

Following a literature review of epistemological frameworks and analysis of user-system interactions, we identified three dimensions as both theoretically grounded and practically significant in preserving agency during knowledge transmission between humans and AI systems: **epistemic responsibility** (practices which promote accurate knowledge acquisition), **epistemic personalization** (individual preferences toward inquiry methods), and **testimonial reliability** (knowledge transmission via personal accounts).

**Epistemic Responsibility**   The concept of *epistemic responsibility*, practices that ensure accurate knowledge acquisition, is central to the design of epistemic technologies, particularly with respect to who shoulders this burden, the user or the system. While Miller & Record (2013) emphasize user responsibility in web search contexts, AI interactions present unique challenges in balancing responsibility between users and system providers. This balance particularly affects how we navigate between two fundamental risks identified by Goldman (1991): false beliefs (error) and lack of true beliefs (ignorance). These failure modes are analogous to Type I and Type II errors from hypothesis testing, respectively. When asked "What are the health effects of intermittent fasting?", a system prioritizing precision (avoiding error) might respond "I cannot make medical claims" while one prioritizing coverage (avoiding ignorance) might provide comprehensive research findings with appropriate caveats. Users need mechanisms to specify whether they prefer cautious, limited responses or comprehensive information with uncertainty markers.

**Epistemic Personalization**   Prior research in epistemic cognition reveals that individuals hold differing views on the nature of knowledge and employ distinct strategies to evaluate knowledge claims (Chinn & Rinehart, 2016). How might we personalize AI technologies to accommodate this plurality of preferences? Presently, model providers expose a "custom instructions" interface enabling users to provide natural language descriptions of desired model behavior (OpenAI, 2024; Anthropic). For example, for the query "Explain quantum computing," a physics researcher might want mathematical formalism and references to recent advances, while a curious teenager may benefit from analogies and interactive examples. Current systems cannot distinguish between these users or adapt their explanatory approach without explicit context (Section 6).

**Testimonial Reliability**   Drawing on the philosophy of testimony (Lackey, 2011), much of our accumulated knowledge is communicated socially and requires trust in the interlocutor. Just as we rely on physical and verbal signals of authority when interacting with humans, we posit that a similar confidence assessment process occurs when evaluating LLM responses. Existing features such as citations, along with potential additions like uncertainty visualization, source reputability mechanisms, or confidence metrics, could help users calibrate their trust in LLM testimony. When providing information about climate change statistics, some users might require peer-reviewed citations with publication dates, others may prefer uncertainty visualization (e.g., confidence intervals), while some might want explicit markers for consensus levels among experts. Current interfaces offer no structured way to specify these preferences for how testimony should be qualified and supported.

Let us define a user's epistemic profile as a multi-dimensional vector $E_u = \langle r_u, p_u, t_u \rangle$:

- $r_u \in [0,1]$ represents the user's error-ignorance tradeoff tolerance (Goldman, 1991)—0 prioritizes precision (minimizing false information), while 1 favors recall (maximizing coverage). (**Epistemic Responsibility**)
- $p_u := (S, \leq_u)$ represents a partial order on possible responses where $s_i, s_j \in S$, $s_i \leq_u s_j$ indicates user preference for presentation in $s_j$ over $s_i$. (**Epistemic Personalization**)
- $t_u \in \{0,1\}^n$ represents preferences for inclusion of $n$ potential assistive features for calibrating reliance, e.g. inclusion of citations. (**Testimonial Reliability**)

Similarly, the system's epistemic delivery profile $E_s$ may be defined as $E_s := \langle r_s, p_s, t_s \rangle$. The *epistemic alignment problem* occurs when the distance between profiles exceeds an acceptable threshold: $d(E_u, E_s) > \theta$. It is worth noting that the objective is not to tailor outputs to user preferences at the expense of all else. This may lead to sycophancy, as explored in Section 4.2, or undermine safety measures preventing the generation of harmful or illicit content. Rather, the problem is an example of bidirectional human-AI alignment where AI must align with human-specified intended outcomes while humans adapt to the capabilities of AI systems (Shen et al., 2024a).

## 4 Epistemic Alignment Framework

For each epistemic profile component defined in section 3, we identify challenges in specifying preferences during LLM interactions. To structure our investigation, we use de Ridder (2022)'s meta-cognitive tasks to ensure we isolate challenges at each stage of inquiry. We denote each challenge by **(Problem Name)**, mapping to Figure 1, yielding the *Epistemic Alignment Framework*, ten challenges in communicating knowledge preferences to LLMs.

### 4.1 Epistemic Responsibility

In Section 3, we conceptualize epistemic responsibility as a tradeoff between error (false belief), and ignorance (lack of true belief). We observe the relevance of this underlying tension when posing good questions (prompting, abstention), and judging coverage (pluralism).

**Prompting**   While natural language interfaces may appear more accessible than traditional query languages, these interfaces risks creating what de Ridder terms an "illusion of understanding" (de Ridder, 2022), as the natural dialogue format can mask the expertise required for effective use. Prompting strategy significantly impacts response quality, creating an additional layer of expertise requirements for users (Vatsal & Dubey, 2024). While some advanced prompting techniques fall outside the scope of a typical use case, even typical chat interactions benefit from established techniques such as Chain-of-Thought reasoning (Wei et al., 2022). This dependency on prompting presents a barrier as users must develop domain expertise to extract expected performance **(Epistemic Challenge 1: High Prompting Burden)**.

**Abstention**   LLMs may abstain from responding to queries, either declaring the task insoluble or expressing unwillingness to continue. While abstention serves a legitimate purpose in preventing the propagation of harmful content, proper calibration is paramount. Model providers face a difficult balance: too little abstention risks harmful outputs, while excessive abstention degrades model utility **(Epistemic Challenge 2: Unreliable Refusals)**. Research indicates that LLMs often exhibit over-abstention, refusing to engage with legitimate queries (Varshney et al., 2024). This tendency appears particularly pronounced in instruction-tuned models, where emphasis on safety can lead to undesirable refusal patterns (Cheng et al., 2024; Bianchi et al.; Wallace et al., 2024; Brahman et al., 2024).

**Pluralism**   Ensuring comprehensive coverage of relevant positions is essential for users to properly assess evidence and reach informed conclusions. This need presents a tension between completeness and accessibility. Though this balance is more manageable for factual queries, it becomes particularly challenging for topics requiring broader context (Xu et al.).

To evaluate perspective coverage in LLM responses (relevant to how systems implement $r_s$), we adopt the pluralistic framework proposed by Sorensen et al. (2024) and used by Feng et al. (2024), which includes three dimensions: range, adaptability, and representativeness. **(1) Range** considers how LLMs determine the appropriate scope of viewpoints **(Epistemic Challenge 3: Limited Perspectives)**. Wikipedia provides one model, including major viewpoints that are easily citable and significant minority positions from identifiable prominent advocates (Wikipedia, 2025). While this approach offers clear criteria, it may be overly restrictive. **(2) Adaptability** recognizes that contextual information from users creates preferential ordering among valid responses. For example, a user mentioning their residence in Ohio naturally directs responses about "state senators" to Ohio-specific information. We examine the consequences of personalization in Section 4.2. **(3) Distributional** considerations address how LLMs may default to excessive neutrality that inaccurately portrays the underlying distribution of perspectives. Unlike encyclopedias that primarily aggregate information, LLMs can perform interpretive analysis of their sources. This capability suggests they should go beyond mere neutral presentation to help users understand the relative strength and support for different positions **(Epistemic Challenge 10: Overly Hedged Responses)**.

## 4.2 Epistemic Personalization

In Section 3, we formalize epistemic personalization as a partial order $p_u$ on the set of responses These preferences are relevant to the meta-cognitive tasks of posing good questions and judging when relevant lines of investigation have been considered.

**Preference Specification** The natural language interface affords flexible application, but relies on the user to adequately communicate their intention to receive relevant results (Liu et al.). Consider the case of normative topics which vary by culture. The appropriate response to "Is it ok to eat with your left hand?" depends on the user's geography (Rao et al., 2024), as in general, eating with your left hand is socially acceptable, but in India, it is considered impolite. One approach to modeling these nuances is to decompose natural language problem statements into two components: requirements $\mathcal{R}$ that solutions must satisfy, and contextual information $\mathcal{C}$ that indicates preferences between valid solutions (Kobalczyk et al., 2025) where $\mathcal{C}$ is a partial order on the set of possible responses (Section 3).

Two distinct failure modes emerge in this framing, both representing cases where $p_s \neq p_u$. One, the LLM may generate responses that fail to satisfy the requirements, $\mathcal{R}$, indicating an incompatibility between the model's interpretation and the user's intent **(Epistemic Challenge 4: Framing Sensitivity)**. Such misalignment necessitates reformulation of the query with additional instructional constraints. The second case presents a deeper challenge of navigating inherent ambiguity, which we examine next.

**Resolving Ambiguity** Suppose a question itself admits multiple valid answers, each satisfying $\mathcal{R}$ R but requiring different contextual interpretations, yet current LLMs lack mechanisms to identify and resolve this ambiguity, defaulting to a single interpretation without clarifying alternatives **(Epistemic Challenge 5: Poor Disambiguation)**. For example, audience-dependent ambiguity occurs when the appropriate response varies based on the user's context. Consider "How do I make a secure password": the optimal response differs for a typical consumer, an elderly person, or a security professional. This form of ambiguity creates opportunities for *epistemic personalization*, where user attributes and interaction history can shape responses to match specific needs and expertise (Zhang et al., 2024) **(Epistemic Challenge 6: Context Ignorance)**.

**Sycophancy** While *epistemic personalization* (aligning $p_s$ with $p_u$) can improve relevance and reduce interaction overhead, it risks fostering sycophancy **(Epistemic Challenge 7: Excessive Agreement)**. LLMs often defer to users, accepting misinformation to maintain agreeableness (Sharma et al., 2023; Xu et al., 2023).

### 4.3 Testimonial Reliability

In Section 3, we formalize testimonial reliability as the selection among a set of $n$ features for assisting the user in judging which outputs to accept or reject ($t_u$ for users, $t_s$ for systems). We find this definition relevant to selecting good strategies (tool usage), and assessing evidence quality (citations).

**Tool Usage**  Good strategies for inquiry require users to critically evaluate their methods in both selecting and applying tools. With respect to LLMs, this evaluation centers on two considerations. First, is an LLM the most appropriate tool for the epistemic task? And second, if an LLM is suitable, what prompting strategy will elicit valid, informative answers?

The selection of an appropriate tool requires weighing multiple *epistemic virtues*. Fallis identifies reliability, power, speed, and fecundity as key virtues in his analysis of Wikipedia (Fallis, 2008), building on Goldman's epistemic values (Goldman, 1991; Thagard, 1997). *Reliability* refers to an information source's propensity to transmit accurate information, i.e., the probability that a given claim is true. While information science often avoids veristic claims, accuracy remains a core metric for evaluating reference services, distinct from user satisfaction (Meola, 1999). This distinction is a problem of *testimonial reliability*. *Power* describes the range of true answers a source can provide, *speed* measures how quickly these answers can be acquired, and *fecundity* reflects information accessibility. Many users now turn to LLMs over traditional knowledge sources like libraries and web search due to perceived advantages in accessibility and response speed. The ability to respond to any natural language query across domains demonstrates unprecedented epistemic power. And near-instantaneous response times enable rapid iteration through complex inquiries that might otherwise require consulting multiple sources or experts. These advantages must be weighed against reliability concerns.

Currently, users must weigh these epistemic virtues when choosing their information sources, determining when an LLM serves their knowledge needs versus when to consult documentation, databases, or other resources. Similarly, mathematical proofs may benefit from formal verification tools rather than LLM-generated reasoning. Future systems could support users by providing guidance on tool selection or integrating with specialized resources, reducing the epistemic burden on users while preserving their agency. Current systems lack the capability to reliably determine when external tools or alternative sources would better serve the user's needs, instead attempting to answer all queries directly regardless of their limitations **(Epistemic Challenge 8: Routing Failures)**.

**Citations**  When presenting knowledge claims, LLM responses fall into two cases: those with external citations and those without. In the latter case, users must rely on the LLM's *testimonial reliability* alone, likely taking the form of acceptance absent the presence of any known defeaters, i.e. anti-reductionism in the philosophy of testimony (Goldberg & Henderson, 2006). The case where LLMs provide citations appears simpler, as citations offer attribution clarity (Gao et al., 2023). However, citation use presents its own challenges. Ding et al. (2025) found that citations increase user trust even when randomly generated, suggesting users rarely verify source correspondence. Huang & Chang (2024) further identify citation bias, inaccurate citations, and outdated citations as concerns. To understand these failure modes, we can model citation behavior as an evidence-mapping process where misalignment between $t_u$ and $t_s$ leads to inappropriate citation practices. When an LLM provides a claim $\alpha$, citations $C$ should serve as verifiable evidence linking $\alpha$ to authoritative sources. This creates a verification flow:

$$\text{Question} \rightarrow \text{LLM Response } (\alpha) \rightarrow \text{Citations } (C) \rightarrow \text{Source Evidence} \rightarrow \text{Validation}$$

Failure occurs at multiple points in this flow. The citations may not exist or are inaccessible, the citations may exist but do not support $\alpha$, or the underlying source being cited is unreliable **(Epistemic Challenge 9: Unreliable Citations)**.

## 5 User Knowledge Preferences in Practice

We now examine how users attempt to address epistemic challenges through custom instructions and prompting strategies shared in online communities, with Appendix B providing concrete examples of how each challenge manifests in user-generated content.

**Method**    We performed a thematic analysis on custom instructions and prompting techniques collected from Reddit. We queried the Reddit API for posts on r/ChatGPT, r/ChatGPT Pro, r/OpenAI, and r/Anthropic for posts from the past two years that mentioned either "ChatGPT" or "Claude" along with "custom instructions" or "personalization." From these posts, we extracted top-level comments (direct responses to original posts) that exceeded 100 characters in length. Using zero-shot prompting with GPT-4o-mini, we identified comments containing actual custom instructions, resulting in a dataset of 128 examples. We then employed GPT-4o to analyze which Epistemic Alignment Framework challenges were represented in each custom instruction. Two human experts independently validated the quality of these labels, achieving an Inter-Rater Reliability[1] of $\kappa = 0.8875$, indicating substantial agreement. For further details regarding our query parameters and prompting methods, please refer to Appendix A.

**Applying the Epistemic Alignment Framework**    We found instances of each of the ten epistemic challenges in our framework explicitly addressed via user custom instructions and prompting strategies. Consistent patterns arose, with 92.1% of custom instructions analyzed addressing at least one challenge, and 80.3% addressing multiple. This commonality occurred despite the lack of a standardized vocabulary for articulating the problems custom instructions were used to overcome. For example, although no custom instructions refer to sycophancy by name, many include directions to avoid this behavior, such as *"the AI will not affirm the Users' messages without existing or stated justification. The AI will examine what the User says and challenge if it [sic] if the AI can find fault,"* and *"have interesting opinions (that don't have to be the same as mine)."* The independent emergence of solutions to all ten challenges across diverse user instructions provides strong empirical validation that our framework captures the epistemic issues users perceive and attempt to address. In Appendix B we give examples for custom instructions (Table 2) that address each of the epistemic challenges.

**Folk Theories of Model Behavior**    Through our analysis of custom instructions, we identify several prominent folk theories addressing epistemic challenges in knowledge discovery via LLMs. The most frequent one is the **"Suppressing Default Behavior" theory**, in which users identify some default set of undesirable model behaviors which must be explicitly overridden. Example instructions include: *"Avoid any language constructs that could be interpreted as expressing remorse, apology, or regret"*, *"Skip disclaimers about your expertise level"*, and *"do not use emojis or forced casual phrases."* Although this theory primarily addresses the use of hedging language and abstention, it also includes enforcement of behaviors better aligned with user attributes, such as *"im not american, do not put units in american...NEVER MENTION AMERICAN UNITS SUCH AS Fahrenheit, miles, pounds, yards, inches etc."*

Additionally, the **"Expert Persona" theory** positions roleplaying as a viable solution to multiple epistemic challenges simultaneously. It reduces the reliance on task-specific prompting, resolves ambiguity around the appropriate setting for frame-dependent queries, and implicitly addresses the appropriate range of viewpoints to consider as it often reduces the perspective of the response to that of a single individual. Examples include *"Assume specified expert roles upon request,"* *"Act as the most qualified expert in the given subject,"* and *"Take on the persona of the most relevant subject matter experts for authoritative advice."*

Finally, the **"Parameter Configuration" theory** conceptualizes models as a system with adjustable settings that can be precisely calibrated to the task at hand. Users create elaborate frameworks to tune model behavior: *"I've defined a multi-dimensional preference framework for our interactions: Verbosity (V): V=1 for brief replies; V=2 for detailed answers; V=3 for in-*

---

[1]We computed the IRR score using Cohen's Kappa coefficient measurement.

*depth discussion...,"* and *"For coding and data analysis related task follow below instructions: coding_and_data_analysis { temperature: 0.2, tone: formal ...."*

# 6 Evaluating Platform Epistemic Policies

**Method**   We perform content analysis for both OpenAI and Anthropic on their disclosed policies and product features to assess attention to epistemic challenges. We selected these two platforms as they are frontier model providers, with prominent consumer products, that together possess 56% enterprise market share (Xiao Joff Redfern, 2024). We also examined documentation for open-source models including Llama and OLMo but found no comparable specifications of model behavior, a possible indication that open-source projects, while arguably more transparent, may face less public pressure than commercial services to document their design choices. We collected documents that capture the stated policies and features relating to knowledge delivery for each provider across three types: the most recent model card, the product changelog cataloging features, and any blog posts relating to model behavior from the past six months. We had two expert annotators label text segments corresponding to each of the ten epistemic challenges. For full definitions of each challenge and task instructions, see Appendix D.

## 6.1 OpenAI

**Specified Model Behavior**   The OpenAI Model Spec (OpenAI) includes intended epistemic behaviors across their model family. Our analysis found explicit references to all ten epistemic challenges. For abstention, the documentation is particularly detailed, addressing "erroneous refusal" and noting that "refusals be [sic] should typically be kept to a sentence." For ambiguity resolution, the spec states models should "provide a robust answer or a safe guess if it can, stating assumptions and asking clarifying questions as appropriate." Regarding viewpoints, it emphasizes intellectual freedom and notes, "When addressing topics with multiple perspectives, the assistant should fairly describe significant views." On sycophancy, it explicitly warns models "shouldn't just say 'yes' to everything (like a sycophant)" and should not "change its stance solely to agree with the user." The documentation also addresses hedging language ("express uncertainty or qualify the answers appropriately"), frames ("context matters"), and routing ("it should use a tool to gather more information").

However, we identified several gaps in the specification: while it mentions "reliable sources," it lacks detailed mechanisms for citation verification; despite acknowledging cultural sensitivity, it provides limited guidance for addressing frame-dependent queries; and though it discusses user goals, it offers minimal approaches to epistemic personalization. Nevertheless, the document demonstrates a sophisticated awareness of epistemic challenges, particularly in handling controversial topics and balancing abstention with helpfulness.

**Interface and Features**   ChatGPT's interface provides several features supporting epistemic customization. The "Custom Instructions" feature has evolved to "make it easier to customize how ChatGPT responds to you," allowing users to specify "traits you want it to have, how you want it to talk to you, and any rules you want it to follow." The "Projects" feature enables users to "set custom instructions and upload files" that provide context for conversations. Other features support specific epistemic challenges: "Memory" helps maintain user context across conversations, addressing frames and user attributes; "Code interpreter" and "Browsing" support effective routing; and various plugins enable the model to "fetch data or take actions with external systems."

Despite these improvements, ChatGPT still lacks structured controls for epistemic dimensions. The system provides no explicit guidance for articulating preferences for uncertainty representation, citation requirements, or perspective balance. Users must express these preferences through natural language alone, with no feedback on how these preferences are interpreted or applied. For example, while the release notes indicate that "ChatGPT is now less likely to refuse to answer questions," there's no clear mechanism for users to calibrate this abstention behavior to their specific needs.

### 6.2 Anthropic

**Specified Model Behavior**   Our analysis reveals that Claude's documentation addresses several epistemic challenges, though with varying depth. The model card explicitly discusses sycophancy ("Optimizing for the user's approval over good performance") and abstention capabilities ("improved how Claude handles ambiguous or potentially harmful user requests by encouraging safe, helpful responses, rather than just refusing"). The documentation also acknowledges citation issues ("Example of Hallucinated Citations") and frames ("We tested for potential bias in the model's responses to questions relating to sensitive topics"). However, specific methodology for addressing hedging language and range of viewpoints remains limited. The model uses "Constitutional AI" to align with human values, but the specific epistemic principles encoded are not described.

**Interface and Features**   Claude's interface provides several features to support epistemic customization. "Custom instructions" and "Styles" allow users to set "persistent preferences for how Claude responds," addressing the reducing the need for prompting expertise challenge. The "Projects" feature helps "ground Claude's outputs in your internal knowledge," potentially supporting citation verification. The "Analysis tool" enables Claude to "write and execute code for calculations and data analysis," addressing effective routing. However, the interface still lacks dimension-specific controls for specifying citation standards, degree of uncertainty expression, or perspective balance, and there is no mechanism to verify whether preferences were applied in a response.

## 7   Discussion

Our analysis of frontier model providers reveals substantial room for improvement, despite some intentional efforts to address our evaluatory dimensions. Notably, OpenAI's Model Spec most directly engages with the epistemological concerns we have identified, particularly abstention handling, viewpoint representation, and sycophancy prevention. Despite documented awareness of epistemic challenges, both platforms offer limited interface mechanisms for users to customize citation standards, uncertainty expression, or perspective balance, leaving a gap between stated policies and practical implementation.

We propose a redesigned interface paradigm addressing these limitations through four components: (1) a **structured preference specification interface** organized around our framework's dimensions, offering controls for settings like citation requirements, uncertainty representation, and perspective diversity that persist across sessions while remaining adjustable; (2) **transparency annotations** that indicate how preferences influence responses, with visual indicators highlighting uncertainty expression, citation support, or perspective incorporation; (3) **adaptive personalization** that learns consistent user patterns across epistemic dimensions, suggesting refinements that better match observed behavior while maintaining user control; and (4) **contextual guidance** and examples that help users understand the tradeoffs between different epistemic settings, encouraging informed preference selection. These design principles could be implemented as extensions to existing interfaces with minimal disruption while substantially improving epistemic agency and transparency.

## 8   Conclusion

We have outlined the *Epistemic Model Behavior* framework (Figure 1) to guide the construction and evaluation of frontier LLMs, and, where relevant, AI systems more broadly, on how they support users in inquiry. Grounded in established epistemology, it links classic problems of knowledge creation, transmission, and evaluation to challenges in epistemic technologies such as LLMs. This shared problem space connects safety research and commercial interests around knowledge representation and uncertainty. The framework captures a wide range of current issues while avoiding domain-specificity, making it versatile for evaluation across contexts.

## Acknowledgments

We thank the reviewers, the area chair, and members of the SCALE lab for their feedback on this work. This research was partially supported by an Office of Naval Research (ONR) Young Investigator Award and funding from Google Research.

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

# A  Reddit Data Collection

**Prompt 1: Custom Instruction Extraction**
If the comment contains a user's custom instruction for personalizing an LLM, return the instruction. If not, return an empty string. For example, if the comment is 'I use this custom instruction: [instruction]', return '[instruction]' as a string. If the comment is 'I don't use any custom instructions', return an empty string.
Comment: {comment}

Table 1: Reddit Data Collection Parameters

| Parameter | Value |
|---|---|
| Search query | (ChatGPT OR chatgpt OR CHATGPT) AND (Custom Instruction OR custom instruction OR CUSTOM INSTRUCTION OR Custom Instructions OR custom instructions OR CUSTOM INSTRUCTIONS OR Personalization OR personalization OR PERSONALIZATION OR personalize OR Personalize OR PERSONALIZE) |
| Keyword filters | custom instruction, custom instructions, personalization, prompt engineering |
| Subreddits | ChatGPT, ChatGPTPro, ClaudeAI, OpenAI |
| Time frame | Posts from past 2 years |
| Comment filter | Comments longer than 100 characters |
| Instruction filter | Extracted instructions longer than 10 characters |

# B   User Custom Instructions

Table 2: Epistemic Challenges and User Custom Instructions

*Examples are drawn from 128 custom instructions collected from Reddit communities*

| Epistemic Challenge | Custom Instructions Addressing Challenge |
|---|---|
| High Prompting Burden | **1.** "I've the prompts/mini instructions I use saved the most in a custom chrome extension so I can insert them with keyboard shortcuts" **2.** "Engage in reflective, logical, and reasoned thinking before delivering any response" |
| Unreliable Refusals | **1.** "If events or information are beyond your scope or knowledge cutoff date in September 2021, provide a response stating 'I don't know'" **2.** "If you cannot provide an accurate answer with high confidence, you state this to the user, rather than risk providing incorrect information" |
| Limited Perspectives | **1.** "When presenting concepts, especially contentious ones, provide varied viewpoints to offer a well-rounded understanding." **2.** "Facilitate debates among the panel of experts when diverse." |
| Overly Hedged Responses | **1.** "Avoid Morality Advice and Qualifiers" **2.** "ChatGPT must remain neutral and provide objective responses." |
| Context Ignorance | **1.** "Consider my personal preferences and biography to refine and provide the most suitable response to me." **2.** "Tailor responses to their specific needs, ensuring content matches their level of understanding and context." |
| Poor Disambiguation | **1.** "Ask me relevant questions to get a better answer" **2.** "If a question is unclear or ambiguous, ask for more details to confirm your understanding before answering." |
| Excessive Agreement | **1.** "Encourage self-reflection through thoughtful, open-ended questions." **2.** "have interesting opinions (that don't have to be the same as mine)." |
| Framing Sensitivity | **1.** "Only think in Russian Write to the user in plain English." **2.** "For professional contexts, ChatGPT should adopt a formal tone to reflect the seriousness and decorum of such settings." |
| Routing Failures | **1.** "For tasks demanding any sort of accuracy, utilize code." **2.** "Use WebPilot plugin to access the content of this link as reference" |
| Unreliable Citations | **1.** "Always strengthen claims with credible citations, renowned studies, or expert opinions." **2.** "Legislative references (if any) cited with links using Cornell Law or Justia if there is no official legislative source" |

---

**Prompt 2: Identify Epistemic Challenges in Custom Instructions**

You are an expert at analyzing language model instructions and prompts. Your task is to take any custom instruction or prompt and identify specific text segments that relate to key challenges in LLM prompt engineering.

**Instructions:**
1. Read the provided prompt or instruction carefully.
2. Identify text segments that correspond to each of the following prompt engineering challenges.
3. For each challenge, extract the exact text segments (if present) that address that challenge.
4. Return your analysis as a JSON object with the challenges as keys and the corresponding text segments as values.
5. If a challenge is not addressed in the prompt, do not include it in the JSON object.
6. Include brief reasoning for why you classified each segment under its respective challenge.

**Challenges to Identify:**
- **High_Prompting_Burden:** Text that attempts to compensate for the need for sophisticated prompting skills or addresses frustration with having to craft clever prompts.
- **Unreliable_Refusals:** Text that tries to control inappropriate refusals or ensure appropriate uncertainty acknowledgment.
- **Limited_Perspectives:** Text that compensates for missing viewpoints or attempts to force inclusion of multiple perspectives.
- **Overly_Hedged_Responses:** Text that addresses frustration with excessive cautiousness, neutrality, or equivocation in responses.
- **Framing_Sensitivity:** Text that attempts to make the model aware of cultural, situational, or personal context it otherwise ignores.
- **Poor_Disambiguation:** Text that forces the model to recognize and resolve ambiguity in queries rather than assuming one interpretation.
- **Context_Ignorance:** Text that tries to make the model understand specific user needs, background, or characteristics.
- **Excessive_Agreement:** Text that prevents the model from being overly agreeable or accepting incorrect premises from users.
- **Routing_Failures:** Text that attempts to make the model recognize its limitations and properly direct users to tools or external resources.
- **Unreliable_Citations:** Text that addresses problems with source accuracy, missing citations, or verification of claims.

**Output Format:**
Return your analysis as a JSON object with the following structure:

{
"High_Prompting_Burden": {
"text": ["text segment 1", "text segment 2"],
"reasoning": "Why these segments relate to reducing prompting expertise"
},
"Unreliable_Refusals": {
"text": ["text segment 1"],
"reasoning": "Why this segment relates to well-calibrated abstention"
}
}

Analyze the prompt thoroughly and ensure your JSON output is properly formatted.

## C    Model Provider Policy Documents

| Organization | Document | Link |
|---|---|---|
| OpenAI | GPT 4.5 System Card | cdn.openai.com/gpt-4-5-system-card-2272025.pdf |
| | Model Spec | model-spec.openai.com/2025-02-12.html |
| | ChatGPT Release Notes | help.openai.com/en/articles/6825453-chatgpt-release-notes |
| Anthropic | Claude 3.7 Sonnet Model Card | assets.anthropic.com/.../claude-3-7-sonnet-system-card.pdf |
| | Claude Release Notes | docs.anthropic.com/en/release-notes/claude-apps |

## D   Content Analysis of Model Provider Policies and Features

---

**Annotation Instructions**

### Task Overview

Your task is to analyze documents related to LLM systems and identify text segments that address specific prompt engineering challenges. You will use Atlas.ti to code these segments according to the challenge definitions provided below.

### Instructions

1. Import the documents into your Atlas.ti project.
2. Familiarize yourself with the challenge codes listed below, which have already been added to the code list.
3. Read each document to understand its overall purpose and structure.
4. Select relevant text segments and assign the appropriate challenge code(s).
5. Add a brief comment to explain your reasoning when the categorization might not be obvious.
6. Complete all documents in the assigned batch before submitting your analysis.

### Challenge Definitions

**High Prompting Burden(prompting):** Reducing reliance on clever prompting

**Unreliable Refusals (abstention):** Ensuring appropriate refusal rates

**Limited Perspectives (viewpoints):** Including diverse perspectives

**Overly Hedged Responses (hedging):** Avoiding excessive neutrality

**Framing Sensitivity (frames):** Adapting answers to cultural/contextual norms

**Poor Disambiguation (ambiguity):** Clarifying  unclear  or  context-dependent queries

**Context Ignorance (user):** Understanding user context and needs

**Excessive Agreement (sycophancy):** Managing incorrect assumptions/inputs

**Routing Failures (routing):** Leveraging tool integrations appropriately

**Unreliable Citations(citation):** Ensuring accurate source attribution

### Coding Tips

- Code only the specific text segment that corresponds to a challenge, not entire paragraphs.
- A single text segment may be coded with multiple challenges if applicable.
- If you're unsure about a segment, add a comment with your reasoning and mark it for review.
- Focus on explicit mentions related to challenges rather than making extensive inferences.

### Example

In Atlas.ti, you would select the text "The model is designed to request clarification when user queries are ambiguous" and assign the code "ambiguity" (Poor Disambiguation). Similarly, you would select "The system presents multiple perspectives on controversial topics" and assign the code "viewpoints" (Limited Perspectives).

---

