# OpenReview forum: "Epistemic Alignment: A Mediating Framework for User-LLM Knowledge Delivery"
_colmweb.org/COLM/2025/Conference — COLM 2025_

### Official Review · Reviewer_yf1j · 2025-04-14

**Rating:** 8
**Confidence:** 3
**Ethics Flag:** 1

**Summary:**

This paper tackles a critical and often overlooked aspect of LLMs development: how these models deliver information to users.
While much recent work in LLM research has focused on hallucinations, safety, and bias, the authors spotlight a different (but related) question: how can LLMs be designed so that they align with the epistemic preferences of individual users?

**Reasons To Accept:**

The paper’s major theoretical contribution is a ten-challenge framework that clarifies mismatches between users’ knowledge-related preferences and typical LLM outputs.
The framework is broken down into ten specific challenges that are derived from both philosophical analysis and real user queries/instructions. The framework seems as both rigorous and immediately useful.

To validate the framework, the authors conduct a thematic analysis of “custom instructions” and prompt-engineering tricks posted online (e.g., Reddit). These real-world examples show that users repeatedly try to solve the same set of problems, despite having no standardized approach to do so. This offers solid empirical grounding for the framework.
The authors review documentation from two leading providers (OpenAI and Anthropic). Although both platforms show awareness of certain epistemic concerns, neither one provides robust, structured ways for users to specify or verify detailed epistemic preferences. Instead, the preferences remain limited to free-form text, which can be inconsistently interpreted by the model.

The paper proposes a new interface paradigm. This includes interface elements to specify citation standards, uncertainty levels, or perspective range, and a more verifiable record of whether the model complied.

The perspective presented in the paper can help designers and researchers move beyond the question “Does the model respond accurately?” to “How well does the model reflect—and adapt to—the user’s unique epistemic goals?”

**Reasons To Reject:**

The authors acknowledge (but do not fully explore) the conflict between personal epistemic alignment and global safety filters. Balancing personalization with guardrails remains tricky. While the paper recommends that personalization not override safety, a deeper exploration of how to implement that boundary in practice would be valuable.

The paper gives only high-level design principles rather than step-by-step guidelines or prototypes.

---

> ### Author Response · Authors · 2025-06-03
>
> We sincerely thank the reviewer for their thoughtful assessment. We're especially grateful for the recognition of our framework's theoretical rigor and practical relevance, as well as the value seen in our empirical validation and interface proposals.
>
> We agree the tension between personalization and safety is critical. In Section 3, we frame this as a bidirectional alignment problem and note risks like sycophancy and safety compromise. Implementing guardrails that respect user preferences without enabling misuse would likely require preference optimization or post-hoc verification layers—both active areas of research. We see our framework as a necessary precursor to such work, clarifying where these tradeoffs are most likely to arise, such as in Section 4.1.
>
> We appreciate the observation concerning high-level design principles versus step-by-step guidelines. Our intention was to establish a conceptual foundation for future work rather than specify implementation details. As discussed in Section 7, we identify concrete interface components derived from our analysis. However, instantiating and evaluating these components would require system development and user testing beyond the scope of this paper. We hope this framework serves as a roadmap for subsequent research and design efforts.

---

> > ### Comment · Reviewer_yf1j · 2025-06-05
> >
> > Thanks for the answer.
> > I keep my rating unchanged (clear accept).

---

### Official Review · Reviewer_EdtK · 2025-05-10

**Rating:** 6
**Confidence:** 4
**Ethics Flag:** 1

**Summary:**

The paper provides a framework for reviewing the capabilities of large language models in relation to user knowledge.

The framework is tested in a qualitative manner by analysing documentation of certain LLM providers, namely openAI and anthropic.

The overall goal of the paper is very valuable. The weakest point is it evaluation against LLM providers. With the dynamics in the generative AI industry, providers change and their offerings as well quite fast. The authors should consider an evaluation against reference data. I understand that this is not in the scope of their current research but would help to generalise their results.

Such data based approach could also give more weight to the recommendations in section 7. For example, values provided in a structured preference specification interface ultimately needs to be processed by the LLM. However, LLMs do not make any sense of preferences, for LLMs these are just additional input tokens. Hence, the quality of preference processing needs to be improved by related training data and evaluated by benchmarking data.

**Reasons To Accept:**

A new framework for defining user knowledge delivery to LLMs.

**Reasons To Reject:**

The evaluation is specific to two LLM providers and hard to replicate and generalise.

---

> ### Author Response · Authors · 2025-06-03
>
> We appreciate the reviewer’s thoughtful comments and support for the framework’s contribution.
>
> We agree that provider documentation is subject to rapid change and that evaluation against a stable reference dataset would improve replicability and generalizability. Our current analysis aimed to assess the extent to which real-world systems support epistemic alignment, but we see clear value in complementing this with reference-based evaluations. As the reviewer suggests, assessing how LLMs interpret structured epistemic preferences would require both training-time support and benchmark datasets to test compliance. We view this as a critical direction for future work and believe our framework can inform the construction of such datasets by identifying the specific challenges to benchmark against.

---

### Official Review · Reviewer_Cydn · 2025-05-13

**Rating:** 7
**Confidence:** 4
**Ethics Flag:** 1

**Summary:**

This paper draws from theories of social epistemology and epistemic cognition and uses them as a principled framework to address how LLMs should present information in a structured interface when interacting with the user, when used as a knowledge acquisition tool. It is written clearly, with detailed outlines of the framework and concrete analysis of existing LLMs and recommendations. The language and jargons/terminologies of the framework can be convoluted and hard to follow at times; but I believe they are able to get the point across. It certainly provides an original way of thinking about how LLMs should present information and knowledge, even though the framework itself is pre-existing and well established in other disciplines that are non technical. It may prove to be of significance in terms of shaping the high level directions that LLMs might take, and governing the evaluation of LLM-like tools at large.

**Reasons To Accept:**

Even though this paper is not a typical technical paper with experiments and novel quantitative methods, I encourage the community to be open minded in accepting this type of interdisciplinary work, as it provides a structured way to think about the high level behaviors of LLMs as a tool of knowledge acquisition that otherwise may be elusive in a rapidly evolving field that may seem chaotic and unscientific at times, which is full of ad-hoc methodologies at analyzing the behaviors of LLMs and governing/evaluating how it should be like.

**Reasons To Reject:**

It is also possible that after all these philosophically inspired theoretical jargon and framework to look at LLMs, there is little practical value and impact in terms of how users will interact with the LLMs - i.e., it largely remains a theoretical construct that has no practical significance.

---

> ### Author Response · Authors · 2025-06-03
>
> We appreciate the reviewer’s recognition of the paper’s originality and conceptual clarity.
>
> While we understand the concern that the framework could remain abstract, other reviewers have pointed to two clear application areas: benchmarking and system design. Both align closely with the framework’s intended purpose.
>
> To support practical implementation, we also outline in Section 7 a set of concrete design components, such as structured preference specification and transparency annotations, that translate the framework into actionable system features. We hope to explore these directions further in future work.
>
> We also appreciate the feedback on language and terminology. Our goal was to maintain conceptual rigor while being accessible, and we will continue refining the language to improve clarity for broader audiences.

---

### Official Review · Reviewer_o8K6 · 2025-05-22

**Rating:** 7
**Confidence:** 3
**Ethics Flag:** 1

**Summary:**

Authors gave a very nice rebuttal. I've revised my score substantially upward from 4 to 7.

---------------------------

This work studies the use of LMs for knowledge acquisition & problems/challenges that arise from users relying on unstructured natural language prompts as primary interface for articulating their knowledge acquisition preferences. The work presents a Framework which is a description of 10 challenges/problems/topics related to 3 core ideas in epistemology. The work is split into parts – first explaining these ideas in epistemology, then describing these challenges in AI with examples. Then in the latter 1/3rd of the paper, shows validity of these characterized challenges/problems by analyzing Reddit shared prompts and user rationales for how those prompts are designed to show that they often are trying to address one or more of these challenges. And finally they do an analysis of OpenAI and Anthropic product documents to show a lack of guidance on how people should better use their models to overcome these challenges.

**Questions To Authors:**

see concerns above

**Reasons To Accept:**

Overall, I love the topic & would welcome this type of work at COLM. The work itself serves as an interesting blend of survey paper + position paper + social science. The study on Reddit shared prompts is a cool idea & serves as good evidence for the problems enumerated in this work. The connection between these problems that the community has been tackling to classical ideas in epistemology - I haven’t seen before; intellectually stimulating.

**Reasons To Reject:**

My main concern is that this paper is quite hard to read through; it feels a bit like an HCI paper that has gone through cuts/revisions but not enough until the work is polished. In summary, my main concerns are: (1) a lack of connection to related works outside of the cited works in epistemology & HCI, (2) assertions for how to best design AI interfaces that appear without sufficient argument/evidence, (3) unclear boundary between what is prior work’s ideas vs contribution of this work, (4) a gap in scope when applying the ideas in this work to a broader set of actors in the field of AI (beyond 2 companies) that would enable more representative claims about the AI ecosystem. I go into more detail before, organized by section of paper:

—

*Related Work*

I found the Related Work lacking. As it is written, it serves more as Background on epistemology. This is fine, but a preferred Related Work would answer the question of how this work’s main problem (i.e. difficulty in users expressing knowledge acquisition preferences in unstructured prompts) & contributions (i.e. the introduction of framework, the validation against shared prompts, the application to proprietary models) are situated relative to other works in the field. For example, after reading Related Work, I want to know what other works have studied this problem (in knowledge-acquisition context, or even generally in other areas related to difficulties in user expression of intent in unstructured prompts), what other works have done to tackle this problem (maybe the field has focused on technical solutions, but lacks frameworks like yours? Or the field has plenty of frameworks, but differs from yours in some way?) What other works have studied this phenomenon of sharing prompts, or conducted similar studies with studying/coding/analysis of user prompts and relating this to how we should better inform designing these LM interfaces? Maybe these types of questions or frameworks like yours have been answered for other technologies but not for LMs specifically?  A Related Work section doesn’t need to tackle all of these comprehensively, but those are just some examples of what I want to get out of Related Work.

To contrast this, what I’m reading now with current Related Work falls short of this. The narrative is perfectly fine & sensible: “Inquiry is important because primary means for acquiring knowledge. But inquiry is tough, especially because of illusions of understanding. Some meta-cognitive tasks (1-5) help combat these illusions. And there’s a ton of variation in strategies because people are different (cultures, disciplines, etc).”  I think this is all perfectly fine, but can’t comprise the sole Related Work section; I’d revise to compress this into a single paragraph & make room for the other unaddressed Related Work questions.


*Problem Definition*

* Epistemic Responsibility. The leap here is a bit far. The passage appears to be first posing the big question as a User vs System has responsibility to ensure accurate knowledge acquisition. But going from there to Error-Ignorance tradeoff is too big a leap. That is, it is not clear how picking any point on Error-Ignorance tradeoff translates to where User vs System should have more responsibility. This passage is confusing.

* Epistemic Personalization & Testimonial Reliability description makes sense, albeit a bit redundant with Intro & Related Work.

* L137-L147 restating the defined 3 dimensions but with math notation - why is this necessary? The math notation doesn’t get used again in rest of paper; it’s not clear what this adds beyond the definitions above in this same section.

*Epistemic Alignment Framework*

* It’s unclear the criteria of what makes it into the “10 challenges”. For example, the 3 top level categories (Epistemic Responsibility, Epistemic Personalization, Testimonial Reliability), I can understand this because it is clear this is borrowing de Ridder (2022)’s tasks. But for these 10 challenges, I can’t tell if these are user challenges, system design challenges, challenges for the AI developer community, or…? Who are these a challenge to? Calling them all challenges is also a bit odd – Sycophancy and Ambiguity seem like challenges, but Abstention and Citation seem more like desired model behaviors (and the research question is more about how to do it well). “Prompting” is especially ambiguous when framed as a “challenge”, unless one means a challenge for users; but then the other challenges like “Tool Usage” aren’t user challenges.

* Idea: Can you rephrase these challenges with verbs? For example, in Table 2 “Reducing Prompting Expertise” is much more useful than just a heading “Prompting”.

* There are points in this section where it is clear what is different coming from this work; example where the paper says “We argue that…”. But these seem like assertions without backing evidence. For example, consider L248-L251 – “We argue that few legacy epistemic institutions are competitive with LLMs in terms of power and speed.”  What is backing this assertion?  Or in L254 - “Currently, the task of selecting appropriate tools rests with users” – where is evidence that this is true? There is a significant amount of work on agents where main handlers are responsible for their own tool use. None of these works are cited/discussed here to support this assertion. L258 – “We argue that this epistemic responsibility can be safely assumed by model providers with minimal infringement on user agency.” – again, based on what evidence; there’s no references or discussion to support this.

*User Knowledge Preferences in Practice*

This part of the work is cool. I like how it uses Reddit comments and user rationales around why prompts are crafted a certain way. One couldn’t have done this analysis on public prompt sets like WildChat alone. I feel like I understood more about the ideas in this work from looking at Prompt 2 and Table 2 together than much of the prose in the first couple sections of the paper. Consider significantly cutting redundant language (see comments below), unrelated related work (see comments above), and using the space to add a table of these examples. Otherwise, overall this is a good section.

*Evaluating Platform Epistemic Policies*

I dislike this motivation for limiting analysis to OpenAI & Anthropic – “We selected these two platforms as they are frontier model providers, with prominent consumer products, that together possess 56% enterprise market share”. Limiting to only these two developers is problematic because as a reader, I want to maximize the value I get from seeing application of this framework used to explain something about the broader AI landscape. And limiting to these two providers misses painting a picture of two large groups of players in AI – open source, and smaller proprietary AI (e.g. startups). This also focuses entirely on “general purpose” models; what about the landscape of domain-specific or application-specific LMs? There is so much more one could do here. This is akin to seeing works where they introduce a cool method for large-scale science-of-science study & they pick only a 1-2 journals, or a cool method for performing large-scale multilingual NLP analysis & showing its use on bilingual corpora. The work just feels incomplete & overly specific to what two companies are doing, as if unaware there is a much larger space of AI happening beyond these two companies.

*Other Writing*

Feels verbose/full of redundancies. This makes the paper’s narrative difficult to follow.
* For example, are L95-L101 really saying anything different from the first sentence in Intro? Are L102-108 saying anything different from the second sentence in the intro? Sure there is more specifics, but that’s two paragraphs dedicated to motivating the study, which I had already bought the premise of after the intro. It’s keeping me from reaching the main contributions of this work.

* Another example - The first occurrence of Chinn & Rinehart 2016 AIR framework citation in Related Work is fine. But it comes up again in Problem Definition making the same point about individuals & different strategies. In fact, the whole paragraph in Related Work on Epistemic Cognition seems to be about making this point around diverse strategies arising from individuals being different in some ways. It feels redundant with the Epistemic Personalization paragraph which makes this point again. Can these be merged?

In the abstract and Introduction, the work was talking about a problem with prompting being “unstructured” natural language; the abstract even calls out “no structured way to articulate these preferences”. But I didn’t see anywhere that discusses why/how structure would improve things? The same thing about assertion in L51 that this work suggests need to “provide transparent feedback about how parameters shape knowledge delivery”. I don’t see where in this paper this is argued/supported. Most of the paper is around presenting background on epistemology (Sec 3), problems/challenges (Sec 4), and then goes into proof of the framework in Sec 5 and application of it in Sec 6. The only exception is L412+ in the Discussion/Conclusion, where the work re-states the proposal for these types of interfaces. But this seems like it comes out of nowhere with a majority of the work not discussing it until the very last paragraph. Is there a missing Design Implications section?

---

> ### Author Response · Authors · 2025-06-03
>
> **Related Work and Contributions**
>
> We appreciate the reviewer’s thoughtful comments on the Related Work section. When revising, we will compress the epistemology discussion into a single paragraph and expand the section to directly address three areas aligned with our core contributions. First, we will add work on **knowledge-acquisition failure modes and frameworks**, including hallucination, factuality, and knowledge conflict (Huang 2023; Wang 2023; Jin 2024; Xu 2024; Gao 2023). These works primarily assess task performance (e.g., hallucinations, factuality), but lack a systematic study of how users engage in knowledge acquisition with LLMs. Second, we will incorporate literature on **expressing user intent in unstructured prompts**, such as ambiguity resolution, intent taxonomies, and prompt rewriting (Min 2020; Mao 2023; Bodonhelyi 2024). These studies focus on ambiguity within general conversations or QA tasks, but do not address the specific factors that shape users’ needs when evaluating AI-generated outputs. Third, we will draw on research into **prompt-sharing practices** across communities such as Midjourney forums and jailbreak subreddits (Goloujeh 2024; Shen 2024; Trippas 2024). We believe this sufficiently situates our paper with respect to topically relevant prior work.
>
> Additionally, we will revise several statements to better distinguish between normative proposals and descriptive claims. The line about LLMs and legacy institutions (L248-L251) was intended to reflect a broad shift in user behavior, not a quantified comparison. On tool use (L254), we were referring to users' strategies for deciding when and how to rely on an LLM for a given task rather than autonomous agents. The suggestion that model providers take on more responsibility (L258) is meant as a forward-looking design implication, not a claim about current deployment. We will amend these lines to minimize ambiguity.
>
> **Problem Definition**
>
> Our goal is to show that error (false beliefs) and ignorance (missing true beliefs) are the two core failure modes in knowledge acquisition, as identified by Goldman (1991), and that the trade-off between them helps frame how epistemic responsibility is distributed. This trade-off can be navigated by either the system or the user, and divergence in these choices may lead to misalignment. For example, a system that prioritizes minimizing error may abstain from uncertain responses, even if the user would prefer broader coverage.
>
> Additionally, we included the mathematical notation to formalize the user and system profiles as a foundation for epistemic alignment. This formalism is used to structure the challenges in Section 4 and inform the design principles discussed in Section 7. For example, the user’s error–ignorance trade-off parameter $r_u$ underpins the “Well-Calibrated Abstention” challenge; the partial order $p_u$ motivates our treatment of “Preference Specification” and “Ambiguity Resolution”; and the binary vector $t_u$ corresponds to assistive features like citation inclusion and uncertainty visualization.
>
> **Epistemic Alignment Framework**
>
> Concerning the choice of the term “challenge,” our intent was to capture issues relating to user experience. We will reword the ten challenges to ensure they are user-centric, for example, changing "Citation Verification" to "Unreliable Citations" and "Reducing Prompting Expertise" to "High Prompting Burden."
>
> **User Knowledge Preferences in Practice**
>
> We appreciate the reviewer’s positive response to our thematic analysis of prompts shared via Reddit. We will include more direct references to Appendix B in Section 5, but are unable to include Table 2 in the main text due to space constraints.
>
> **Evaluating Platform Epistemic Policies**
>
> We agree that applying the framework to a wider range of models and settings, especially beyond general-purpose systems, is an important direction. As part of this work, we looked at documentation for open-source models such as Meta’s Llama and AI2’s OLMo and did not find comparable discussions of epistemic responsibilities or behaviors. So far, it seems that only the most mature and commercially established models have begun to adopt this kind of language. This paper aims to lay the groundwork for broader adoption by offering both a vocabulary and a set of recommendations. We will clarify this point in the paper and may include a brief note on other models we reviewed to reinforce this observation.
>
> **Other Writing**
>
> We will condense the epistemological framing in Section 2 into a more compact summary.
>
> The concern about unstructured prompting and lack of transparency is primarily what motivates our analysis of Reddit user workarounds, which show repeated attempts to resolve the same underlying challenges, albeit informally and inconsistently (e.g., L305-27). We take this pattern as evidence that users and platforms are already grappling with the need to represent epistemic preferences more explicitly.

---

> > ### Author Response · Authors · 2025-06-03
> > **References**
> >
> > - **Bodonhelyi, A.**, Bozkir, E., Yang, S., Kasneci, E., & Kasneci, G. (2024).
> >   *User intent recognition and satisfaction with large language models: A user study with ChatGPT.*
> >   arXiv preprint [arXiv:2402.02136](https://arxiv.org/abs/2402.02136)
> >
> > - **Gao, T.**, Yen, H., Yu, J., & Chen, D. (2023).
> >   *Enabling large language models to generate text with citations.*
> >   arXiv preprint [arXiv:2305.14627](https://arxiv.org/abs/2305.14627)
> >
> > - **Huang, L.**, Yu, W., Ma, W., Zhong, W., Feng, Z., Wang, H., ... & Liu, T. (2025).
> >   *A survey on hallucination in large language models: Principles, taxonomy, challenges, and open questions.*
> >   *ACM Transactions on Information Systems, 43*(2), 1–55.
> >
> > - **Jin, Z.**, Cao, P., Chen, Y., Liu, K., Jiang, X., Xu, J., ... & Zhao, J. (2024).
> >   *Tug-of-war between knowledge: Exploring and resolving knowledge conflicts in retrieval-augmented language models.*
> >   arXiv preprint [arXiv:2402.14409](https://arxiv.org/abs/2402.14409)
> >
> > - **Mahdavi Goloujeh, A.**, Sullivan, A., & Magerko, B. (2024).
> >   *The Social Construction of Generative AI Prompts.*
> >   In *Extended Abstracts of the CHI Conference on Human Factors in Computing Systems*. Association for Computing Machinery.
> >
> > - **Mao, K.**, Dou, Z., Mo, F., Hou, J., Chen, H., & Qian, H. (2023).
> >   *Large language models know your contextual search intent: A prompting framework for conversational search.*
> >   arXiv preprint [arXiv:2303.06573](https://arxiv.org/abs/2303.06573)
> >
> > - **Min, S.**, Michael, J., Hajishirzi, H., & Zettlemoyer, L. (2020).
> >   *AmbigQA: Answering ambiguous open-domain questions.*
> >   arXiv preprint [arXiv:2004.10645](https://arxiv.org/abs/2004.10645)
> >
> > - **Wang, Y.**, Feng, S., Wang, H., Shi, W., Balachandran, V., He, T., & Tsvetkov, Y. (2023).
> >   *Resolving knowledge conflicts in large language models.*
> >   arXiv preprint [arXiv:2310.00935](https://arxiv.org/abs/2310.00935)
> >
> > - **Shen, X.**, Chen, Z., Backes, M., Shen, Y., & Zhang, Y. (2024, December).
> >   *"Do anything now": Characterizing and evaluating in-the-wild jailbreak prompts on large language models.*
> >   In *Proceedings of the 2024 ACM SIGSAC Conference on Computer and Communications Security* (pp. 1671–1685).
> >
> > - **Trippas, J.**, Al Lawati, S., Mackenzie, J., & Gallagher, L. (2024).
> >   *What do users really ask large language models? An initial log analysis of Google Bard interactions in the wild.*
> >   In *Proceedings of the 47th International ACM SIGIR Conference on Research and Development in Information Retrieval* (pp. 2703–2707). Association for Computing Machinery.
> >
> > - **Xu, R.**, Qi, Z., Guo, Z., Wang, C., Wang, H., Zhang, Y., & Xu, W. (2024).
> >   *Knowledge conflicts for LLMs: A survey.*
> >   arXiv preprint [arXiv:2403.08319](https://arxiv.org/abs/2403.08319)

---

> > ### Comment · Reviewer_o8K6 · 2025-06-10
> >
> > Thanks! Your rebuttal has cleared up a lot for me, and I'll revise my score accordingly.
> >
> > **Related Work and Contributions**
> > Thanks, I think making the revisions you are stating above will really help contextualize the work for COLM audience.
> > As for some of the other points, those are my misunderstandings when reading your paper. If there is a way you can clarify language a bit for future readers who may have similar misunderstandings, that'd be much appreciated.
> >
> > **Problem Definition**
> > Honestly, this is what made it click for me in your rebuttal:
> > >For example, a system that prioritizes minimizing error may abstain from uncertain responses, even if the user would prefer broader coverage.
> > The prose in this section (L115-L122) is quite high level. Even your explanation in rebuttal feels like restating the same thing, up until this example. I feel like adding even a sentence of example for each of these areas--Epistemic responsibility, Epistemic Personalization, Testimonial Reliability--could *really* help! That is, don't just approach framework top-down with high level definitions, but also bottom up with some examples.
> >
> > Again, on math notation, this is where it clicked for me:
> > >for example, the user’s error–ignorance trade-off parameter underpins the “Well-Calibrated Abstention” challenge; the partial order motivates our treatment of “Preference Specification” and “Ambiguity Resolution”; and the binary vector corresponds to assistive features like citation inclusion and uncertainty visualization.
> > I still believe if you introduce math notation, that it should be used more than the place it's introduced. So like in your example here, I'd like to see a sentence of example in the paragraph on "Well Calibrated Abstention" that uses $r_u$ explicitly. And so on for the other passages as well.
> >
> > **Epistemic Alignment Framework**
> > Thanks for these changes!
> >
> > **User Knowledge Preferences in Practice**
> > This would be great, thanks!
> >
> > **Evaluating Platform Epistemic Policies**
> > Thanks, even just including a sentence or two about reviewing other models & finding them lacking comparable discussions would be good. On the whole, it does bring up the question of -- why don't open source models provide this? Does it even make sense to expect this of open source models given the way they aren't deployed with lay users in mind but released artifacts to be adapted for use by other developers? Or is it merely oversight from these developers and they really should be doing it? Not a requirement, but would be interesting to see some thought on this, given so much attention on AI ecosystem now is on open source vs closed source, and ensuring your framework is relevant to open source somehow can broaden the audience for your work.
> >
> > **Other Writing**
> > That would be helpful, thanks!

---

### Decision · Program_Chairs · 2025-07-08

**Decision:**

Accept

**Comment:**

The reviewers agree that this is high quality work. They particularly appreciate the integration of clearly-introduced philosophical concepts with proposals for concrete practical interventions, as well as empirical research demonstrating that these epistemological concepts adequately describe real-world user experiences.

Reviewers made some suggestions for improving the clarity of the piece; authors have proposed some responses in their rebuttal that adequately address these concerns.

The work is clearly original—as reviewer Cydn observes it brings together a range of disciplinary perspectives in a productive way, to address an unaddressed problem. This is also clarified in the proposed additions to the 'related work' section that o8K6 recommended, to which the authors responded well.

Reviewers also agree that this work is significant. Epistemic alignment is important; the empirical research shows this is not just a theoretical problem but one that is actually experienced by users. Reviewers did recommend broader application beyond the two model providers on which the work's practical implications focus, but the authors addressed that concern well in their rebuttal.

Reviewers did indicate some modest concerns, but on the whole were very positive about the paper.